# Impaired Humoral Immunity with Concomitant Preserved T Cell Reactivity in IBD Patients on Treatment with Infliximab 6 Month after Vaccination with the SARS-CoV-2 mRNA Vaccine BNT162b2: A Pilot Study

**DOI:** 10.3390/jpm12050694

**Published:** 2022-04-26

**Authors:** Richard Vollenberg, Phil-Robin Tepasse, Eva Lorentzen, Tobias Max Nowacki

**Affiliations:** 1Department of Medicine B for Gastroenterology, Hepatology, Endocrinology and Clincial Infectiology, University Hospital Muenster, 48149 Muenster, Germany; phil-robin.tepasse@ukmuenster.de (P.-R.T.); tobias.nowacki@ukmuenster.de (T.M.N.); 2Institute of Virology, University Hospital Muenster, 48149 Muenster, Germany; eva.lorentzen@ukmuenster.de; 3Department of Medicine, Gastroenterology, Marienhospital Steinfurt, 48565 Steinfurt, Germany

**Keywords:** SARS-CoV-2, COVID-19, vaccination, IBD patients, seroconversion, humoral and T-cellular immune response

## Abstract

Introduction: The Coronavirus Disease 2019 (COVID-19) pandemic has been caused by the severe acute respiratory syndrome coronavirus-2 (SARS-CoV-2). The most important approach to prevent severe disease progression and to contain the pandemic is the use of COVID-19 vaccines. The aim of this study was to investigate the humoral and cellular response in immunosuppressed patients with inflammatory bowel disease (IBD) on treatment with anti-TNF (infliximab, adalimumab) and anti-α4ß7-Integrin (vedolizumab) 6 months after mRNA vaccination against SARS-CoV-2 compared to healthy subjects. Methods: In this prospective study, 20 IBD patients and 9 healthy controls were included 6 months after the second BNT162b2 vaccination. In addition to quantitative determination of IgG antibody levels against the SARS-CoV-2 receptor-binding domain (RBD) of the spike protein subunit S1, a SARS-CoV-2 surrogate neutralization test (sVNT) was used to assess potential neutralization capacity. SARS-CoV-2-specific T-cell responses were measured using an interferon-γ (IFN-γ) release assay (IGRA; Euroimmun Medical Laboratory Diagnostics, Lübeck, Germany). Results: S-IgG could still be detected in the majority of IBD patients 6 months after second vaccination. Compared to healthy controls, IBD patients treated with anti-TNF agents showed both lower neutralizing activity in sVNT (percent inhibition of ACE2 receptor binding by RBD protein) and lower IgG-S (AU/mL) antibody levels (AB) (sVNT: 79% vs. 2%, *p* < 0. 001; AB: 1018 AU/mL vs. 141 AU/mL, *p* = 0.025). In contrast, patients on therapy with vedolizumab showed no impairment in humoral immune response (sVNT, S-IgG) compared with healthy controls. Specific T-cellular reactivity was detected in 73% of IBD patients and in 67% of healthy controls independent of immunosuppressive therapy (anti-TNF., vedolizumab) (*p* = 0.189). Conclusion: Six months after BNT162b2 vaccination, this study found significantly decreased antibody levels in patients under anti-TNF therapy. IBD patients under anti-TNF and vedolizumab therapy had no impairment of T-cellular reactivity compared to healthy controls at this time point. Further studies with larger collectives for confirmation should follow.

## 1. Introduction

Severe acute respiratory syndrome coronavirus-2 (SARS-CoV-2) is the pathogen of the Coronavirus Disease 2019 (COVID-19). The disease was first described in Wuhan, China, in December 2019 and has spread worldwide from there, causing the COVID-19 pandemic [1]. While COVID-19 results in mild to moderate symptoms in most people, it triggers severe illness with acute respiratory distress syndrome (ARDS) in a subset of patients [2,3,4]. Critical courses of disease are associated with severe lung injury, multiple organ failure, and high mortality. In severe COVID-19 courses, signs of hyperinflammation, as in classic cytokine storm syndromes, have been described [5,6]. Even after mild courses of the disease, some patients suffer from persistent symptoms, a symptom complex summarized as “long-COVID-19” [7,8,9]. Ulcerative colitis and Crohn’s disease are inflammatory bowel diseases (IBDs). For drug therapy for remission induction and remission maintenance, various immunosuppressive drugs are used to reduce intestinal inflammatory activity [10,11]. Immunosuppressive therapies (e.g., monoclonal antibodies or azathioprine) can lead to serious side effects, such as opportunistic infections. Since immunosuppressed patients have a higher susceptibility to severe disease progression in infections with endemic pathogens [12], the COVID-19 pandemic has raised major concerns about immunosuppressive therapy in IBD patients. Based on a possibly altered immunological status in IBD patients and depending on the existing immunosuppressive therapy, recent studies have indicated a potential association with severe disease progression [13]. Therapeutic approaches are being developed worldwide to combat COVID-19. In the European Union (EU) and Germany, in addition to anti-inflammatory and antiviral drugs, monoclonal antibodies were approved for passive immunization after SARS-CoV-2 infection [14]. Currently, the most effective protection against symptomatic SARS-CoV-2 infection and against severe disease progression can be achieved by vaccination. Therefore, vaccination is considered the most appropriate approach to combat the COVID-19 pandemic. In Europe, the mRNA vaccines mRNA-1273 and BNT162b2, the vector vaccines Ad26.CoV2.S and ChAdOx1, and the protein-based vaccine NVX-CoV2373 are currently approved [15,16,17,18]. The pivotal studies have demonstrated high efficacy of these vaccines in preventing severe and critical courses of the disease in immunocompetent patients. Patients with compromised immune systems due to pre-existing conditions or medication were not included in the pivotal studies. Recent studies have shown an attenuated immune response in IBD patients three months after vaccination depending on existing immunomodulatory therapy.

In a previous study, we found an impaired humoral immune response 3 months after second vaccination with an mRNA vaccine, particularly in patients on anti-TNF therapy, compared with healthy control subjects. Six months after the second vaccination, we found, in a cohort of only 4 IBD patients and 7 healthy controls, the first indications of a significantly more divergent humoral immune response in IBD patients compared to healthy controls [19].

We therefore conclude that it is still unclear as to what extent long-term humoral response after SARS-CoV-2 vaccination is impacted by immunosuppressive therapies [19,20,21,22,23,24,25,26,27,28].

The aim of this study was to investigate the humoral response in a larger IBD patient collective supplemented by the investigation of T-cellular reactivity six months after mRNA vaccination against COVID-19 compared to healthy subjects.

## 2. Methods

### 2.1. Study Subjects and Samples

The study was conducted in a prospective, monocentric study design. Serum samples were collected from IBD patients (*n* = 20) and healthy controls (*n* = 9) in the IBD outpatient clinic of the Department of Gastroenterology, Hepatology, Endocrinology, and Clinical Infectiology at Münster University Hospital, Germany, from June 2021 to March 2022. Vaccination of patients and controls was performed independently of this study on the recommendation of the treating physician. Blood samples were collected 6 months after the second vaccination (±20 days). Healthy controls had no previous diseases and were not taking regular medication. All included patients and healthy controls were asked about suspected or antigen/PCR test-proven SARS-CoV-2 infection. In addition, sera were tested for anti-nucleocapsid SARS-CoV-2 IgG as a marker of SARS-CoV-2 infection. Patients with proven/suspected COVID-19 disease (positive PCR or rapid antigen test, borderline/positive anti-nucleocapsid SARS-CoV-2 IgG test) were excluded from the study (*n* = 1 healthy controls). Only patients and controls vaccinated with the BNT162b2 mRNA vaccine were included in the study. For IBD patients with ulcerative colitis, the Mayo clinical score was determined at the time of blood collection, and for Crohn’s disease patients, the Crohn’s disease activity index (CDAI score) was determined in each case. According to the immunosuppressive therapy (adalimumab, infliximab, vedolizumab, azathioprine), IBD patients were divided into subgroups. All included patients had no change in therapy since the time point 12 weeks before the first vaccination (Figure 1). The study was approved by the local ethics committee (Münster University Hospital: 2021-039-f-S), and the study participants gave written consent.

### 2.2. Quantification of Serum Markers

As an indicator of previous SARS-CoV-2 infection, qualitative IgG antibody determination against the nucleocapsid protein (N) of SARS-CoV-2 (SARS-CoV-2 IgG assay, Abbott Diagnostics, Abbott Park, North Chicago, IL, USA) was performed. IgG antibodies against the SARS-CoV-2 receptor binding domain (RBD) of the spike protein subunit S1 were quantified as a sign of a performed COVID-19 vaccination or/and a previous infection (SARS-CoV-2 IgG II Quant Assay, Abbott Diagnostics). The assays based on the chemiluminescence microparticle immunoassay (CMIA) technique are certified for clinical use (CE/IVD). Diagnostics were performed according to the manufacturer’s instructions (ARCHITECT device, Abbott, Chicago, IL, USA). All results were given as relative light units (rlu). Values for IgG (N) were calculated as indices of sample rlu divided by calibrator rlu (S/C), where S/C indices below 0.49 were considered negative, S/C indices between 0.49 and 1.39 were considered borderline, and S/C indices of 1.4 and above were considered positive. RBD IgG (S-IgG, quantitative) were determined from data reduction curves of sample and calibrator values (AU/mL). Values greater than/equal to 50.0 AU/mL indicate seropositivity. The cPassTM SARS-CoV-2 Neutralization Antibody Detection Kit (GenScript Biotech, Mainz, Germany), also CE/IVD certified, was used to assess the potential neutralization capacity. This surrogate virus neutralization assay quantifies inhibition of RBD protein binding to the human host cell receptor protein ACE2 by patient antibodies in a blocking ELISA format and correlates with virus neutralization assays [29]. According to the manufacturer’s instructions, patient samples were diluted 10-fold and measured in duplicate. Inhibition was calculated as follows: 1 − (OD value of sample/OD value of negative control) × 100%. Values less than 30% were considered negative; values at or above the cut-off indicated the presence of the SARS-CoV-2 neutralizing antibody. SARS-CoV-2 specific T-cell responses were measured using an interferon-γ (IFN-γ) release assay (IGRA; Euroimmun Medical Laboratory Diagnostics) according to the manufacturer instructions [30]. For this purpose, S1 peptide-stimulated T cells were measured in whole blood according to the manufacturer’s instructions [31].

### 2.3. SARS-CoV-2 Vaccines

IBD patients and healthy controls received the mRNA vaccine BNT162b2 (BioNTech/Pfizer, New York City, NY, USA), which is approved in the European Union (EU) for vaccination against SARS-CoV-2. The patients received the vaccine independently of this study on the recommendation of their treating physicians. Patients and controls vaccinated with the adenovirus vector-based vaccines Ad26.CoV2.S (Johnson & Johnson, New Brunswick, NJ, USA) and/or ChAdOx1 (AstraZeneca, Cambridge, UK), or the mRNA vaccine mRNA-1273 (Moderna, Cambridge, MA, USA), were excluded from the study [15,16,17,18].

### 2.4. Statistical Analysis

Categorical variables (representation of absolute numbers, percentages) were compared using the chi-square test or Fisher’s exact test. Continuous variables (presented as medians with interquartile ranges (IQR)) were compared with the *t*-test if normally distributed, and with the Mann–Whitney U test (Wilcoxon) if not normally distributed. For comparison of more than two groups, the Kruskal–Wallis test was performed. To compare subgroups, the Bonferroni correction post hoc test was performed when variance was equal (Levene’s test), and the Games–Howell test was performed when variance was different. Multicomparison analyses were performed using GraphPad Prism software (version 8.0 for Microsoft, GraphPad Software, La Jolla, CA, USA). All tests were two-sided, and a *p*-value < 0.05 was considered to indicate a statistically significant difference. All statistical analyses were performed using SPSS 26 (IBM, Chicago, IL, USA).

## 3. Results

### 3.1. Cohort Characteristics

Of the 20 recruited IBD patients and 9 healthy controls, all individuals with suspected or confirmed SARS-CoV-2 infection were excluded from the study (*n* = 0 IBD patients, *n* = 1 healthy controls). As inclusion criteria, only patients and controls vaccinated twice with the mRNA vaccine BNT162b2 were included. The mean age of IBD patients was 42 years, and 55% were male. The body mass index (BMI) was 23 (22–24) kg/m^2^. There were no significant differences between patients and controls with respect to these characteristics (Table 1). Crohn’s disease was present in 45% of IBD patients (median CDAI score 0 [0]), and 55% of patients had known ulcerative colitis (median Mayo score 2 [0–3.5]). Oral mesalazine therapy was given in 20% of patients and oral predisolone therapy in 10%. Mesalazine therapy (supp.) was administered in 10%. Budesonide therapy (oral/supp.) was not used in any of the patients. The most common concomitant diseases in the patients were pre-existing cardiovascular disease (5%) and diabetes mellitus (5%) (Table 1). IBD patients were subdivided according to their current immunosuppressive therapies (*n* = 4 vedolizumab, *n* = 11 anti-TNF (infliximab/adalimumab), *n* = 2 azathioprine + anti-TNF). With regard to patient characteristics (age, sex, and BMI), there were no significant differences between the subgroups of IBD patients. There were no significant differences between groups in terms of known pre-existing conditions.

### 3.2. Significantly Decreased SARS-CoV-2 S-IgG and sVNT Inhibition Levels in IBD Patients 6 Months after Second Vaccination

SARS-CoV-2 S-IgG (AU/mL) and sVNT (percent inhibition) results were compared between all IBD patients and healthy controls six months after the second vaccination. IBD patients exhibited significantly decreased sVNT inhibition levels and S-IgG antibody levels (sVNT 14% (0–52%) vs. 79% (57–85%), *p* = 0.003; S-IgG 189 AU/mL (22–514 AU/mL) vs. 1018 AU/mL (618–1583 AU/mL), *p* = 0.002) (Table 2). In subgroup analysis between IBD patients with existing immunosuppressive therapies (vedolizumab, anti-TNF, anti-TNF + azathioprine), there were no significant differences in S-IgG levels or sVNT levels 6 months after the second vaccination (Post-Hoc-Tests: *p* > 0.05) (Figure 2a,b). In a subgroup analysis of IBD patients compared to healthy controls, significantly reduced sVNT and S-IgG levels were detected in patients on immunosuppressive therapy with anti-TNF agents (sVNT: 2.4% (0–43%) vs. 79% (57–85%), *p* < 0.001; S-IgG: 142 AU/mL (0–340 AU/mL) vs. 1017 AU/mL (618–1584 AU/mL), *p* = 0.025) and anti-TNF + azathioprine (sVNT: 0% (0–0%) vs. 79% (57–85%), *p* < 0.001; S-IgG: 98 AU/mL (98–98 AU/mL) vs. 1017 AU/mL (618–1584 AU/mL), *p* = 0.009). IBD patients on therapy with vedolizumab did not show reduced S-IgG and sVNT levels compared to healthy controls (sVNT: 55% (31–87%) vs. 79% (57–85%), *p* > 0.05; S-IgG: 694 AU/mL (370–3633 AU/mL) vs. 1017 AU/mL (618–1584 AU/mL), *p* > 0.05).

### 3.3. Reduced Seroconversion Rates in IBD Patients Six Months after SARS-CoV-2 mRNA-Vaccination

Six months after the second COVID-19 vaccination, significantly lower seroconversion rates according to sVNT (inhibition > 30%) were observed in the immunocompromised IBD patients compared to the healthy controls (45% vs. 100%, *p* = 0.005). In contrast, SARS-CoV-2-S IgG levels (seropositivity > 49 AU/mL) 6 months after the second vaccination did not show significantly higher conversion rates in healthy controls (100%) compared to IBD patients (75%, *p* = 0.153). Depending on the existing immunosuppressive therapy (vedolizumab, anti-TNF, anti-TNF + azathioprine), there were no significant differences with respect to seroconversion rates (sVNT, S-IgG) (Figure 3a,b).

### 3.4. Sustained T-Cellular Reactivity in Immunosuppressed IBD Patients 6 Months after Second Vaccination

There were no significant differences between immunosuppressed IBD patients and healthy controls with respect to T-cellular reactivity six months after second vaccination against SARS-CoV-2. Positive T-cell reactivity was detectable in 73% of IBD patients compared to 67% of healthy controls (*p* = 0.189). Only marginal T-cell reactivity was detectable in 27% of patients (controls: 13%, *p =* 0.189). In the subgroup analysis of IBD patients after existing immunosuppression, there were no significant differences (*p* > 0.05) (Figure 4).

## 4. Discussion

Previous studies have examined the protective efficacy following dual SARS-CoV-2 vaccination in immunocompromised IBD patients in most studies at an interval of up to 12 weeks after complete baseline immunization. In these studies, S-IgG seroconversion rates and antibody levels were primarily considered. Regarding T-cellular reactivity after vaccination, there are only few data available, and these are furthermore limited to the first weeks after immunization. To our knowledge, the present study is the first to examine seroconversion rates, SARS-CoV-2 S-IgG and sVNT levels, and T-cellular reactivity in IBD patients compared with healthy subjects 6 months after the second vaccination with BNT162b2.

In a previous study, we found, in a cohort of only 4 IBD patients and 7 healthy controls, the first indications of a significantly more divergent humoral immune response in IBD patients compared to healthy controls 6 months after the second vaccination [19]. The study revealed significantly different absolute antibody levels and neutralization rates by sVNT in IBD patients compared with healthy subjects, but statistical significance may not have been reached due to low patient numbers [19].

In the present study, IBD patients receiving immunosuppressive therapy had significantly lower S-IgG levels (*p* = 0.001) than healthy controls six months after the second vaccination, but independent of individual existing immunosuppressive therapies, our study found no significant differences of S-IgG seroconversion rates in IBD patients (75% vs. 100% in healthy controls) 6 months after vaccination. These data support the findings of Frey et al. (2022) [32], who demonstrated a seroconversion rate (S-IgG) in 78.7% of immunosuppressed IBD patients 6 months after COVID-19 vaccination. Nevertheless, the sVNT assay in our study revealed significantly different seroconversion rates between both groups (45% in IBD patients vs. 100% in healthy controls) and significantly different percentage sVNT inhibition values (sVNT 14% in IBD patients vs. 79% in healthy controls, *p* = 0.003). Finally, besides reduced S-IgG levels, our data indicate a reduced neutralizing capacity of antibodies in IBD patients compared with healthy individuals 6 months after vaccination. To investigate effects of individual immunosuppressive therapies on neutralizing capacity, studies in larger cohorts are needed.

Going deeper into subgroup analysis, IBD patients treated with anti-TNF agents (infliximab/adalimumab) or anti-TNF agents and azathioprine had significantly lower S-IgG/sVNT levels than healthy controls. In contrast, IBD patients on therapy with vedolizumab and healthy controls did not differ significantly (*p* > 0.05) with respect to S-IgG and sVNT inhibition levels. However, our study corroborates previous studies that demonstrated significantly reduced antibody levels in immunocompromised IBD patients compared to healthy controls in the 2–12-week period after second vaccination [28,33,34,35,36]. In particular, patients on anti-TNF therapy showed significantly reduced sVNT inhibitory levels and decreased S-IgG levels [33,34,37,38,39]. Shebab et al. (2021) and Doherty et al. (2022) already demonstrated reduced seroconversion rates in patients on anti-TNF medication [35,40]. Patients on vedolizumab did not show impaired humoral immune responses up to 12 weeks after second vaccination compared to healthy controls [40,41].

In contrast, patients treated with vedolizumab were unaffected in their humoral immunity compared to healthy controls, which can be explained by the mechanism of action in targeting a gut-specific anti-integrin that does not impair systemic immunity and is in line with previous findings [42]. Furthermore, our results confirm previous studies showing attenuated vaccine efficacy in IBD patients on immunosuppressive therapies after vaccination against various other pathogens. In particular, patients on therapy with the immunomodulators methotrexate and thiopurines have been shown to have an attenuated serologic response to influenza vaccines, hepatitis B vaccines, and pneumococcal vaccines [41,43,44]. However, anti-TNF antibodies also lead to a reduced immune response and vaccine efficacy after a single hepatitis A or B vaccination in IBD patients [45,46]. Other studies have described impaired influenza and pneumococcal vaccination in IBD patients receiving therapy with anti-TNF [46]. Reduced responses after hepatitis A are also described in patients with other diseases such as rheumatoid arthritis on anti-TNF therapy [46]. In comparison, no impairment of immune responses after influenza, measles, mumps, rubella, or hepatitis B vaccination was observed in IBD patients receiving vedolizumab therapy. A possible explanation is the selective mechanism of action of the antibody against the α4β7-integrin, which is mainly expressed in the intestine [47].

At present, there is insufficient knowledge about T-cell reactivity in immunosuppressed IBD patients after SARS-CoV-2 vaccination. Surprisingly, 6 months after the second SARS-CoV-2 vaccination, our study found no differences in T-cellular reactivity between immunosuppressed IBD patients and healthy controls. Subgroup analyses provided no evidence of an influence of individual immunosuppressive therapies (Anti-TNF agents or vedolizumab) on T-cell reactivity. Positive T-cell reactivity was detected in 73% of IBD patients (67% controls) and borderline T-cell reactivity in 27% of IBD patients (13% controls), both results showed no statistical significant differences. Reuken et al. (2021) have demonstrated T-cellular reactivity comparable to healthy controls after the first COVID-19 vaccination in IBD patients independent of existing immunosuppressive therapy [25]. In the study recently published by Li et al., robust T-cell reactivity was demonstrated in IBD patients on anti-TNF therapy 8 weeks after vaccination [48]. According to X et al., T-cellular reactivity was detected in approximately 80% of IBD patients receiving therapy with anti-TNF or vedolizumab at 8 to 10 weeks after COVID-19 secondary inoculation [42]. Our study was now the first to demonstrate comparable T-cell reactivity in IBD patients on immunosuppressive therapy and healthy individuals even 6 months after second SARS-CoV-2 vaccination. These results emphasize the relevance of cellular immunity in preventing severe disease progression, especially considering the markedly reduced humoral immunity in IBD patients shown here and in other studies. Although our study showed a markedly reduced humoral immune response and reduced neutralization capacity in patients receiving anti-TNF therapy, even in these patients a preserved and not reduced T-cellular reactivity could be detected compared to healthy individuals.

This study has some limitations. While serological markers have been shown to roughly correlate with vaccination success in terms of protective immunity, a quantitative threshold could not be identified [49]. Hence, we characterized the potential neutralizing activity of patient sera by both a quantitative anti-spike IgG assay and sVNT [29,50]. Cutoff values given by manufacturers reflect diagnostic criteria for positivity while not warranting protection against infection [49]. Some of the IBD patients had other pre-existing conditions (cardiovascular disease, diabetes) compared to healthy controls. The impact of preexisting conditions on the efficacy of vaccination against SARS-CoV-2 does not appear to be clear. Although Naschitz et al. showed attenuated humoral responses in people with diabetes, cancer, and multiple diseases [51], a more recent study found comparable humoral responses in patients with type I and type II diabetes compared with healthy controls, but showed attenuated responses in renal insufficiency [52]. In our study, T-cell reactivity was tested using only a SARS quantiferon assay with determination of the amount of interferon-gamma released after stimulation with SARS-CoV-2 specific proteins. These results cannot provide a fully comprehensive test of T-cell immunity, because T-cell immunity is subject to very complex mechanisms apart from interferon-gamma synthesis. In this study, a small patient and control population was investigated. In particular, the results of the subgroup analyses have limited statistical power as a consequence. Further studies with larger collectives should follow.

## 5. Conclusions

In our study, we demonstrated for the first time significantly reduced S-IgG levels and sVNT inhibition levels in IBD patients on anti-TNF therapy 6 months after the second SARS-CoV-2 vaccination. In contrast, the humoral vaccination response in patients receiving vedolizumab therapy was not affected. Although our study demonstrated a significantly reduced humoral immune response in patients receiving anti-TNF therapy, SARS-CoV-2 specific T-cell reactivity was unaffected in these patients. Our study thus once again emphasizes the importance of cellular immunity and points to the need for individually tailored vaccination regimens, especially for patients on anti-TNF therapy, to optimize the humoral vaccination response. Further studies with larger collectives should follow to confirm these results.

## Figures and Tables

**Figure 1 jpm-12-00694-f001:**
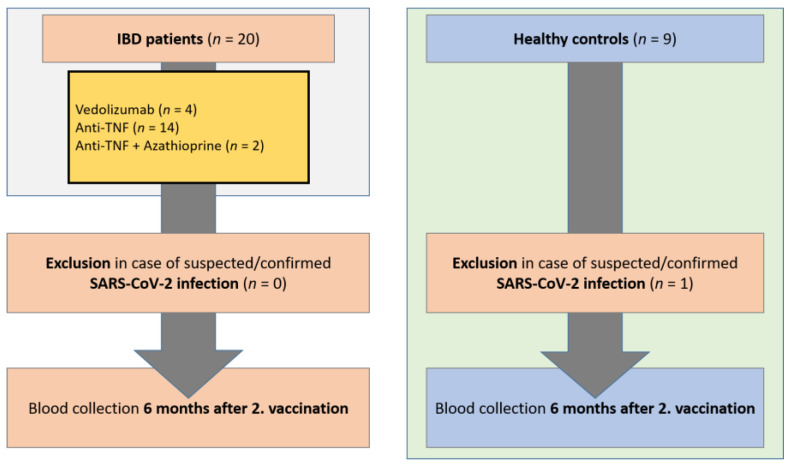
Study flow chart. Inclusion of IBD patients in the period from January 2021 to March 2022 on immunosuppressive medication and healthy controls. Blood collection 6 months following the 2nd vaccination (+/−20 days). Exclusion of patients and controls with suspected or confirmed SARS-CoV-2 infection. IBD, inflammatory bowel disease; SARS-CoV-2, severe acute respiratory syndrome coronavirus-2.

**Figure 2 jpm-12-00694-f002:**
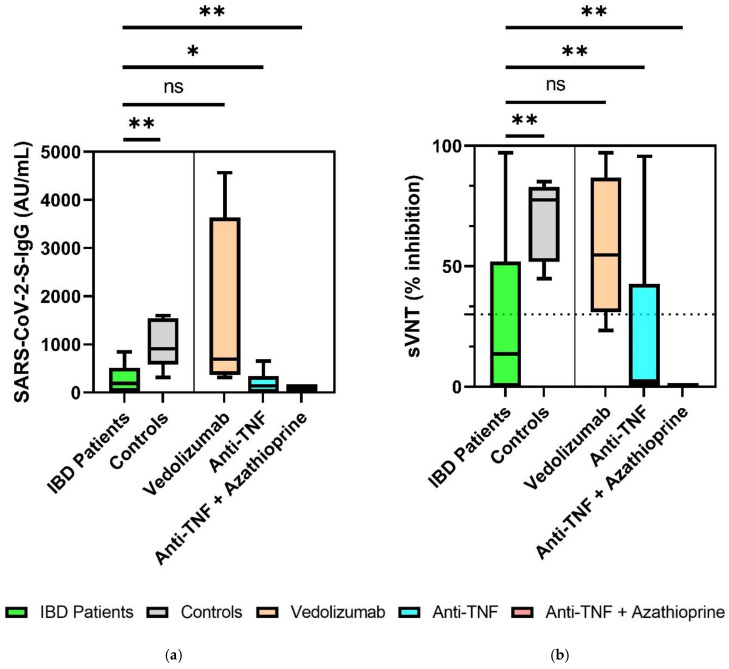
Comparison of SARS-CoV-2-S IgG levels between IBD patients on immunosuppressive therapy and healthy controls 6 months after the second vaccination (**a**) and of sVNT inhibition levels in IBD patient subgroups and healthy controls 6 months after the second vaccination, with the representation of the cut off value (30% inhibition) noted as a dashed line (**b**). * *p <* 0.05, ** *p* < 0.01. ns, not significant. Continuous variables were compared with the *t*-test if normally distributed, and with the Mann–Whitney *U* test (Wilcoxon) if not normally distributed. For comparison of more than two groups, the Kruskal–Wallis test was performed. To compare subgroups, the Bonferroni correction post hoc test was performed when the variance was equal (Levene’s test), and the Games–Howell test was performed when variance was different.

**Figure 3 jpm-12-00694-f003:**
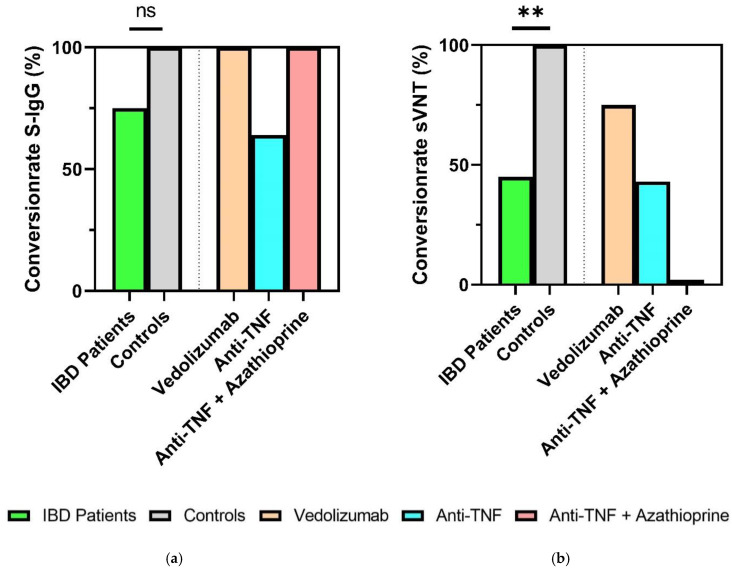
Humoral response of IBD patients and healthy controls 6 months after second vaccination. (**a**) Percentage seroconversion rates (SARS-CoV-2 S-IgG ≥ 50 AU/mL); (**b**) percentage sVNT inhibition. IBD: inflammatory bowel disease. ** *p* < 0.01. ns, not significant. Categorical variables were compared using the chi-square test or Fisher’s exact test.

**Figure 4 jpm-12-00694-f004:**
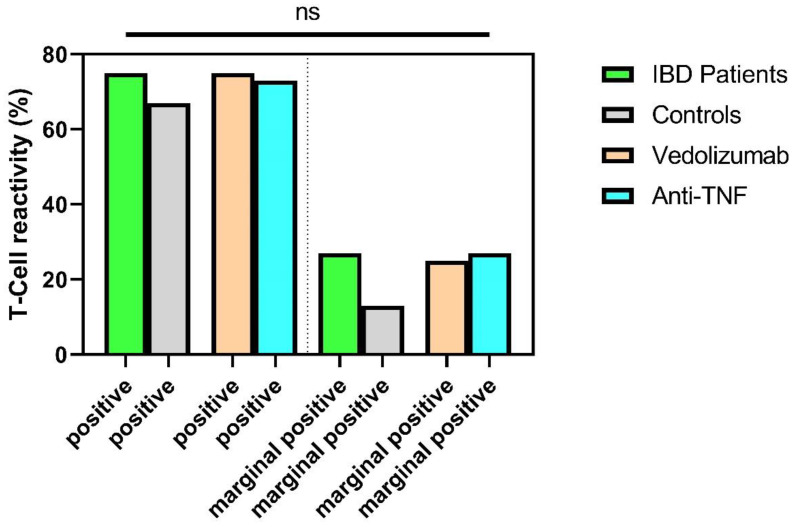
SARS-CoV-2-specific T-cellular reactivity (in%) of IBD patients and healthy controls six months after the second vaccination. Subdivision into positive T-cellular reactivity and marginally positive T cellular reactivity. ns, not significant. Categorical variables were compared using the chi-square test or Fisher’s exact test.

**Table 1 jpm-12-00694-t001:** Cohort characteristics of IBD patients. Representation of the entire IBD patient cohort and classification of patients by immunosuppressive therapy: vedolizumab, Anti-TNF (adalimumab, infliximab), Anti-TNF + azathioprine. IBD: inflammatory bowel disease; IQR: interquartile range; BMI: body mass index; CDAI score: Crohn’s disease activity index; *p*-value ^1^: Mann–Whitney *U* test (Wilcoxon); *p*-value ^2^: Kruskal–Wallis.

Patients		Controls (*n* = 9)	IBD (*n* = 20)	*p*-Value ^1^	Vedolizumab (*n* = 4)	Anti-TNF (*n* = 14)	Anti-TNF + Azathioprine (*n* = 2)	*p*-Value ^2^
Patient characteristics	Age, years median (IQR)	42 (37–57)	39 (26–47)	0.216	40 (35–55)	36 (24–49)	39	0.709
	Sex, male (%)	5 (55)	12 (60)	0.822	4 (100)	8 (57)	2 (100)	0.057
	BMI (kg/m^2^)	23 (22–24)	25 (23–28)	0.198	23 (21–31)	25 (23–28)	27	0.613
IBD	Crohn’s disease (%)	0 (0)	9 (45)		2 (50)	4 (29)	2 (100)	0.634
	CDAI score, median (IQR)	0 (0)	0 (0–0)		120 (0–120)	0 (0–0)	0 (0–0)	0.104
	Ulcerative colitis (%)	0 (0)	11 (55)		2 (50)	10 (71)	0 (0)	0.634
	Mayo score, median (IQR)	0 (0)	2 (0–3.5)		2.5 (2–2.5)	0 (0–0)	0 (0–0)	0.554
Medication	Prednisolone p.o. (%)	0 (0)	2 (10)		2 (50)	0 (0)	0 (0)	0.012
	Budesonide p.o. (%)	0 (0)	0 (0)		0 (0)	0 (0)	0 (0)	1.000
	Budesonide supp. (%)	0 (0)	0 (0)		0 (0)	0 (0)	0 (0)	1.000
	Mesalazine p.o. (%)	0 (0)	4 (20)		0 (0)	4 (29)	0 (0)	0.159
	Mesalazine supp. (%)	0 (0)	2 (10)		0 (0)	2 (14)	0 (0)	0.360
Pre-existing conditions	Cardiovascular disease	0 (0)	1 (5)		1 (25)	0 (0)	0 (0)	0.086
	Respiratory disease (%)	0 (0)	0 (0)		0 (0)	0 (0)	0 (0)	1.000
	Kidney insufficiency (%)	0 (0)	0 (0)		0 (0)	0 (0)	0 (0)	1.000
	Metastatic neoplasm (%)	0 (0)	0 (0)		0 (0)	0 (0)	0 (0)	1.000
	Diabetes (%)	0 (0)	1 (5)		1 (25)	0 (0)	0 (0)	0.086
	Hematologic malignancy (%)	0 (0)	0 (0)		0 (0)	0 (0)	0 (0)	1.000

**Table 2 jpm-12-00694-t002:** T-cell reactivity, SARS-CoV-2 S-IgG levels, and sVNT values of IBD patients and healthy controls 6 months after the second vaccination. Partitioning of the IBD patient population was done based on immunosuppressive therapy: vedolizumab, Anti-TNF (adalimumab, infliximab), Anti-TNF + azathioprine. IBD: inflammatory bowel disease; IQR: interquartile range; *p*-value ^1^: Mann–Whitney *U* test (Wilcoxon); *p*-value ^2^: Kruskal–Wallis.

T-Cell-Reactivity	IBD Patients (*n* = 15)	Controls (*n* = 8)	*p*-Value ^1^	Vedolizumab (*n* = 4)	Anti-TNF (*n* = 11)	Anti-TNF + Azathioprine (*n* = 0)	*p*-Value ^2^
Positive T-Cell reactivity (%)	11 (73)	6 (67)	0.189	3 (75)	8 (73)	0 (0)	1.000
Marginal T-Cell reactivity (%)	4 (27)	1 (13)	0.189	1 (25)	3 (27)	0 (0)	1.000
Negative T-Cell reactivity (%)	0 (0)	1 (13)	0.189	0 (0)	0 (0)	0 (0)	1.000
**Humoral Reactivity**	**IBD Patients** **(*n* = 20)**	**Controls** (***n* = 8)**	***p*-Value ^1^**	**Vedolizumab** **(*n* = 4)**	**Anti-TNF** **(*n* = 14)**	**Anti-TNF + Azathioprine** **(*n* = 2)**	***p*-Value ^2^**
SARS-CoV-2 S-IgG (AU/mL), median (IQR)	189 (22–514)	1018 (618–1583)	0.001	694 (370–3633)	141 (0–340)	98	0.057
Seroconversion rate S-IgG (%)	75	100	0.153	100	64	100	0.640
sVNT (%), median (IQR)	14 (0–52)	79 (57–85)	0.002	55 (31–87)	2 (0–42)	0	0.042
Seroconversion rate sVNT (%)	45	100	0.005	75	43	0	0.088

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
