# Peer review of "Impaired Humoral Immunity with Concomitant Preserved T Cell Reactivity in IBD Patients on Treatment with Infliximab 6 Month after Vaccination with the SARS-CoV-2 mRNA Vaccine BNT162b2: A Pilot Study"

_jpm, 2022, doi:10.3390/jpm12050694_

Round 1
Reviewer 1 Report
Text editing is needed, please remove the following Klicken Sie hier, um Text einzugeben.. The
Sample size is too low, and even less to allow for subanalysis.
Was there a sample size calculation?
Were patients offered an informed consent?
Were similar results with these therapies observed for other immunisations for different disease?
Tables and figures are missing statistical test used in footnote
Spearman rank test may be more appropriate than Pearson correlation for this set of data.
Author Response
Dear editor,
thank you for your letter and your interest in us submitting a revised version of our manuscript, "Impaired humoral immunity with concomitant preserved T cell reactivity in IBD patients on treatment with Infliximab 6 month after vaccination with the SARS-CoV-2 mRNA vaccine BNT162b2: A pilot study". We have addressed each aspect of your letter and all the reviewers’ concerns point-by-point, as indicated below. Relevant changes made in the manuscript are highlighted using red text.
We believe that we addressed all the issues to your satisfaction and want to thank the editor and the reviewers for their efforts to further improve our manuscript.

Reviewer 2 Report
The article “Impaired humoral immunity with concomitant preserved T cell reactivity in IBD patients on treatment with Infliximab 6 months after vaccination with the SARS-CoV-2 mRNA vaccine BNT162b2” investigates the humoral and cellular response in immunosuppressed patients with inflammatory bowel disease on treatment with anti-TNF (infliximab, adalimumab) and anti-α4ß7-Integrin (vedolizumab) 6 months after mRNA vaccination against SARS-CoV-2 compared to healthy subjects. Indeed, it is an interesting study and the authors have presented the data in a good manner. The results of this study indicate a significant decrease in antibody levels in patients under anti-TNF therapy, six months after BNT162b2 vaccination. I have some minor comments to improve this manuscript.
Comments
- Authors can include/discuss a few recent investigations which are missed in the text.
- The number of study samples could have been more.
- The title mentioned in the article and that on the submission site seem to be different.
- There is no description of the author's contribution.
Author Response

(The authors gave the same response as above.)
